# Evaluation of novel glucose-related blood biomarkers for predicting in-hospital mortality in patients with acute ischemic stroke

**Peerapong Kamjai[1], Pornpimon Angkasekwinai**[iD][2,3*]

**1** Department of Medical Technology and Clinical Pathology, Saraburi Hospital, Saraburi, Thailand,
**2** Department of Medical Technology, Faculty of Allied Health Sciences, Thammasat University, Pathum Thani, Thailand, **3** Research Unit in Molecular Pathogenesis and Immunology of Infectious Diseases, Thammasat University, Pathum Thani, Thailand

* upornpim@tu.ac.th, p.akswn@gmail.com

## Abstract

Acute Ischemic Stroke (AIS) is a major cause of death and disability worldwide. AIS patients with hyperglycemia demonstrate a potential risk to enhance severity and mortality rates. Glycemic markers, including blood glucose, hemoglobin A1c, and stress hyperglycemia ratio (SHR), have currently been reported to predict unfavorable outcomes in these patients. However, the ability of novel glucose-related blood biomarkers, such as the glucose to albumin ratio (GAR), glucose to estimated average glucose ratio (GAGR), and glucose to potassium ratio (GPR), to predict severe AIS patients and in-hospital mortality in Thailand remains unclear. This study aimed to investigate the utility of novel glucose related-blood biomarkers in predicting severity and in-hospital mortality among AIS patients. We conducted a retrospective single-center analysis of data from patients admitted to the Stroke Unit at Saraburi Hospital between January 1 and December 31, 2023. A total of 351 AIS patients were examined, with 191 (54.4%) presenting severe cases, and 31 (8.8%) died in the hospital. We demonstrated that the GAR was superior to SHR, GAGR, and GPR in predicting severity, showing an area under the curve (AUC) of 0.672 (95% CI: 0.614–0.731), yielding a sensitivity of 72.8% and a specificity of 56.6%. However, the SHR showed a highest AUC of 0.832 (95% CI: 0.734–0.930) in predicting in-hospital mortality, with sensitivity and specificity of 87.1% and 64.7%, respectively. Furthermore, AIS patients with GAR ≥ 30.0 and SHR ≥ 18.0 had a 12.761 and 12.365-fold increased risk of death (p < 0.001), respectively. Our study indicates that, besides SHR, GAR may serve as a predictive and cost-effective biomarker for predicting severe cases and in-hospital mortality of AIS, facilitating early triage even with limited resources.

**Data availability statement:** All relevant data are within the manuscript and its Supporting Information files.

**Funding:** The author(s) received no specific funding for this work.

**Competing interests:** The authors have declared that no competing interests exist.

## Introduction

Stroke is a significant neurological disorder and a major cause of death and disability worldwide [1]. In Thailand, the prevalence of stroke among adults aged 45 years and older is estimated to be 1.88%, leading to over 50,000 deaths each year [2,3]. The damage of stroke can be classified into two types; ischemic stroke and hemorrhagic stroke [4,5]. Acute ischemic stroke (AIS) is the most common, accounting for 85% of all cases [6]. Approximately 20% of AIS patients present at the hospital with high mortality and poor prognosis, posing a global challenge for clinical management [1,6].

Hyperglycemia is an important risk factor for stroke [7]. Elevated blood glucose levels, observed in about 40–50% of stroke patients, can exacerbate ischemic injury and worsen clinical outcomes [8]. Stress hyperglycemia, a relatively transient increase in blood glucose in response to systemic inflammation or neurohormonal disorders [9], has been correlated to more severe stroke outcomes [10,11], including an increased risk of functional impairment, stroke recurrence, and death. The stress hyperglycemia ratio (SHR), which can be measured as fasting plasma glucose to glycosylated hemoglobin (FPG/HbA1c), has emerged as a promising biomarker for predicting the severity and mortality of AIS patients [12,13].

The stress hyperglycemia ratio (SHR) presently exhibits notable limitations in the early assessment of disease severity, especially in resource-constrained settings with hemoglobin A1c testing [11–13]. Besides SHR, the novel glucose-related blood biomarkers, such as the fasting blood glucose to estimated average glucose ratio (GAGR) [12], glucose to albumin ratio (GAR) [14], glucose to potassium ratio (GPR) [15], and glucose-triglyceride index (TyG) [16], have been used to assess severity and predict mortality rate in AIS patients. Because many patients can progress to critical stages with hemorrhagic transformation [17], fatal outcomes [18], and different therapeutic management [6], an accessible and reliable predictor is crucial for early prognosis and reducing AIS mortality. While SHR has been shown to be a major predictor of fatal ischemic stroke outcomes in Thailand [19], the predictive capacity of novel glucose-related blood biomarkers remains unassessed. This study aimed to determine the usefulness of novel glucose-related blood biomarkers in predicting severity and in-hospital mortality for AIS patients. Our findings may provide insights into prognosis and help reduce the in-hospital mortality rate among AIS patients in Thailand.

## Materials and methods

### Study design and participants

This retrospective single-center study enrolled 351 patients with acute ischemic stroke admitted to the Stroke Unit at Saraburi Hospital in Thailand between 1 January and 31 December 2023. We included patients with AIS aged ≥ 18 years, who arrived within 4.5 hours of symptom onset to the emergency department, diagnosed with acute anterior circulation ischemic stroke by a neurologist and defined under the International Classification of Diseases, 10th revision code for I63 in the electronic medical records. All patients had a routine laboratory examination, including fasting

plasma or blood glucose, hemoglobin A1c (HbA1c), lipid profile, blood urea nitrogen (BUN), creatinine, liver function test, and electrolyte at admission baseline. Patients having transient ischemic attack, minor stroke, posterior circulation stroke, intracranial hemorrhage stroke, pregnancy, refer to other hospital or conditions affecting their blood parameters such as transfusion and fluid replacement, were excluded. The patient recruiting process was illustrated in the patient flow chart (Fig 1).

## Data collection

Our study retrieved data from the electronic medical records of the Stroke Unit at Saraburi Hospital, including demographics, gender, underlying diseases, clinical outcomes, medical treatments, and laboratory findings of the enrolled cases. Patients were categorized into in-hospital mortality (IHM) and survivors, and further classified into mild and severe cases. Severe cases were defined by the National Institutes of Health Stroke Scale (NIHSS) with score > 5 and diagnosed by stroke-certified neurologists, including experiencing complications such as early neurological deterioration (END), requiring oxygen support, and transfer to the intensive care unit. Routine laboratory data at admission baseline included blood glucose and biochemistry parameters. Blood biochemistry tests such as fasting plasma glucose (FPG), lipid profile (cholesterol, triglyceride, HDL, LDL-c), blood urea nitrogen (BUN), creatinine with estimated glomerular filtration rate (eGFR), albumin, and electrolytes (sodium, potassium, carbon dioxide, bicarbonate, anion gap) were performed using an Automated Chemistry Analyzer (Beckman Coulter DxC 700 AU). Hemoglobin A1c (HbA1c) was analyzed using a Mindray BS-820M Automated Analyzer, and estimated average glucose (EAG) was calculated using the formula: $(28.7 \times HbA1c\%)$

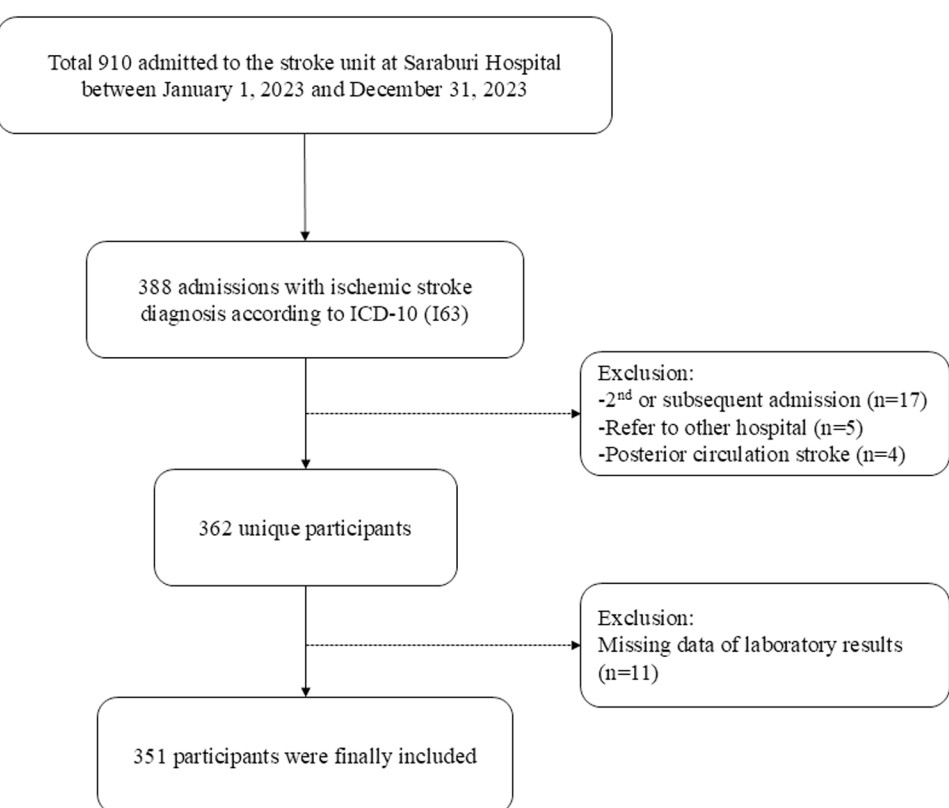

**Fig 1. Patient flow chart.**

– 46.7 (mg/dL). Novel glucose-related blood biomarkers were derived using the following ratios: stress hyperglycemia ratio (SHR) = FPG/ HbA1c, glucose to estimated average glucose ratio (GAGR) = FPG/ EAG, glucose to albumin ratio (GAR) = FPG/ Albumin, glucose to potassium ratio (GPR) = FPG/ Potassium, glucose-blood urea nitrogen to eGFR ratio (GUFR) = (FPG×BUN)/ eGFR, and triglyceride-glucose index (TyG) = Triglyceride×FPG.

## Ethics

The data of this study were fully anonymized, with no identifiable patient information, including hospital numbers, admission numbers, identity card numbers, birthdates, and addresses. This retrospective cohort study was approved by the institutional research ethics committee of Saraburi Hospital (SRBR66–005, EC017/2567) in Thailand. The requirement for informed consent was waived due to minimum risk to the patient and no identifiable information.

## Statistical analysis

The analysis of this study was conducted using the IBM SPSS Statistics version 20.0 statistical package program (Chicago, IL, USA) and GraphPad Prism software version 8.0 (San Diego, CA, USA). The categorical variables, including gender, underlying diseases, and treatments were analyzed by Chi-square or Fisher's exact test, providing the number of units (n), percentage (%) and median. The compliance of the continuous covariate was performed using independent t-tests and the Mann-Whitney U test or log-rank test, respectively. A one-way analysis of variance (ANOVA) or the Kruskal-Wallis's test was used to assess the differences in the median of multiple groups. The examination of a threshold to differentiate severity and mortality groups was performed by receiver operating characteristic (ROC) curves. The cutoff value was determined and chosen by Youden's index based on the appropriate sensitivity and specificity from ROC curve analysis. $P < 0.05$ value was considered statistically significant with 95% confidence interval (95% CI). The multivariate regression model was used to determine odds ratios (ORs) and 95% confidence intervals (CIs) for predicting in-hospital mortality. Variables for adjustment in the multivariate regression model were selected based on prior rationale and known associated risk factors. For in-hospital mortality (IHM), the crude model represents univariable analysis. Model A was adjusted for age, sex, blood pressure, and length of hospital stay. Model B included adjustments for age, NIHSS score, and mRS score. Additionally, model C was further adjusted for age, hypertension, and random blood glucose levels to evaluate novel glucose-related blood biomarkers and their association with fatal outcomes.

## Results

### Baseline characteristics of AIS patients

A total of 351 patients with acute ischemic stroke (AIS) were enrolled in this study. Among cases, 31 AIS patients (8.8%) died in the hospital (in-hospital mortality). The in-hospital mortality rate was associated with the patients' clinical characteristics, including age, gender, preexisting conditions, and treatment complications, as shown in Table 1. The median age of patients who died in the hospital (71±9 years) and those who survived (64±13 years) differed significantly. Among the 351 AIS patients, 314 (89.5%) had underlying conditions. The three most common preexisting conditions were hypertension (78.1%), hyperlipidemia (57.8%), and diabetes (29.6%), respectively. All patients were evaluated for the length of hospital stay. The length of stay in deceased patients was found to exceed 8 days, while it was shorter than 3.5 days in survived cases. Moreover, our study revealed that 53 patients (15.1%) experienced a recurrent stroke and clinical complications during treatment, including stroke-associated pneumonia (SAP) (11.4%), atrial fibrillation (AF) (11.1%), hemorrhagic transformation (3.4%), and sepsis (2.8%), respectively. All AIS patients were treated according to medical practice guidelines and the severity of their condition. Of those patients, 81.7% were given antiplatelet drugs like aspirin, 14.0% were given intravenous thrombolysis with recombinant tissue plasminogen activator (rt-PA), and 9.7% had endovascular thrombectomy (Table 1).

**Table 1. Clinical characteristics in AIS patients.**

| Characteristics | All (n) | IHM | Survivor | p-value |
|---|---|---|---|---|
| Number of cases (%) | 351 | 31 (8.8%) | 320 (91.2%) | |
| Age, mean ± SD | 65 ± 13 | 71 ± 9 | 64 ± 13 | 0.004 |
| Sex, n (%) | | | | |
| Male | 209 (59.5%) | 13 (41.9%) | 196 (61.3%) | 0.036 |
| Female | 142 (40.5%) | 18 (58.1%) | 124 (38.8%) | 0.036 |
| Length of stay, median (IQR) | 4 (3,6) | 8 (3,16) | 3.5 (3,6) | <0.001 |
| Baseline NIHSS score, median (IQR) | 4 (2,9) | 17 (11,21) | 4 (2,8) | <0.001 |
| Discharge mRS score, median (IQR) | 2 (1,4) | 6 (6) | 2 (1,4) | <0.001 |
| **Admission blood pressure (mmHg), mean ± SD** | | | | |
| Systolic blood pressure | 158 (144,177) | 152 (138,177) | 158.5 (145,177) | 0.333 |
| Diastolic blood pressure | 91 (81,101) | 88 (79,95) | 91 (82,101.8) | 0.100 |
| **Underlying disease, n (%)** | | | | |
| No known | 37 (10.5%) | 1 (3.2%) | 36 (11.2%) | 0.227 |
| Hypertension | 274 (78.1%) | 27 (87.1%) | 247 (77.2%) | 0.203 |
| Diabetes | 104 (29.6%) | 12 (38.7%) | 92 (28.7%) | 0.246 |
| Dyslipidemia | 203 (57.8%) | 18 (58.1%) | 185 (57.8%) | 0.978 |
| Cerebrovascular accident | 53 (15.1%) | 8 (25.8%) | 45 (14.1%) | 0.110 |
| Gout | 11 (3.1%) | 2 (6.5%) | 9 (2.8%) | 0.252 |
| Chronic kidney disease | 19 (5.4%) | 4 (12.9%) | 15 (4.7%) | 0.075 |
| Other (cancer, infection, COPD, heart disease and etc.) | 66 (18.8%) | 12 (38.7%) | 54 (16.9%) | 0.003 |
| **Complication, n (%)** | | | | |
| Hemorrhagic transformation | 12 (3.4%) | 5 (16.1%) | 7 (2.2%) | 0.002 |
| Atrial fibrillation | 39 (11.1%) | 5 (16.1%) | 34 (10.6%) | 0.366 |
| Pneumonia | 40 (11.4%) | 23 (74.2%) | 17 (5.3%) | <0.001 |
| Sepsis | 10 (2.8%) | 7 (22.6%) | 3 (0.9%) | <0.001 |
| Urinary tract infection | 11 (3.1%) | 3 (9.7%) | 8 (2.5%) | 0.063 |
| GI bleeding | 5 (1.4%) | 0 (0.0%) | 5 (1.6%) | 0.628 |
| **Treatment, n (%)** | | | | |
| Anti-platelet drug | 287 (81.8%) | 9 (29.0%) | 278 (86.9%) | <0.001 |
| Oral anticoagulants | 34 (9.7%) | 1 (3.2%) | 33 (10.3%) | 0.338 |
| rtPA injection | 49 (14.0%) | 8 (25.8%) | 41 (12.8%) | 0.057 |
| Endovascular thrombectomy | 34 (9.7%) | 8 (25.8%) | 26 (8.1%) | 0.005 |
| Mechanical ventilator | 22 (6.3%) | 16 (51.6%) | 6 (1.9%) | <0.001 |
| Palliative care | 17 (4.8%) | 9 (29.0%) | 8 (2.5%) | <0.001 |

Abbreviations: NIHSS, National Institutes of Health Stroke Scale; mRS, Modified Rankin Score; COPD, Chronic obstructive pulmonary disease; GI bleeding, Gastrointestinal bleeding; rtPA, Recombinant tissue plasminogen activator; IHM, In-hospital mortality; IQR, Interquartile range.

**The routine blood biochemistry and novel glucose-related blood biomarkers in AIS patients.**

The levels of various blood biochemistry are compared between the group of patients who died in the hospital and those who survived and presented in Table 2. The results revealed no significant differences between the groups in terms of hemoglobin A1c, estimated average glucose, blood urea nitrogen, creatinine, sodium, potassium, chloride, triglycerides, and HDL levels. Indeed, the levels of random blood glucose, fasting plasma glucose, estimated glomerular filtration rate (eGFR), carbon dioxide ($CO_2$), anion gap, and LDL-C exhibited statistically significant differences. In deceased cases, the median random blood glucose was 149 mg/dL (range: 131–198 mg/dL) and the fasting plasma glucose was 153 mg/dL

**Table 2. The routine blood biochemistry in AIS patients categorized by in-hospital mortality and survivor groups.**

| Variables | IHM (n = 31) | Survivor (n = 320) | p-value |
|---|---|---|---|
| Random blood glucose at admission (mg/dL), median (IQR) | 149 (131,198) | 98 (106,149) | <0.001 |
| Fasting plasma glucose (mg/dL), median (IQR) | 153 (115,189) | 98 (87,121) | <0.001 |
| Hemoglobin A1c level (%), median (IQR) | 6.0 (5.4,6.7) | 5.8 (5.4,6.4) | 0.438 |
| EAG (mg/dL), median (IQR) | 125 (107,145) | 119 (108.3,137.8) | 0.437 |
| Blood urea nitrogen (mg/dL), median (IQR) | 14.4 (10.2,22.5) | 13.1 (10.4,16.4) | 0.197 |
| Serum creatinine (mg/dL), median (IQR) | 0.97 (0.70,1.33) | 0.86 (0.72,1.06) | 0.181 |
| eGFR (ml/min/1.73 m$^2$) | 65.27 (44.86,91.72) | 86.76 (67.75,96.83) | 0.005 |
| Serum albumin (g/dL), median (IQR) | 3.7 (3.3,4.1) | 3.9 (3.6,4.2) | 0.006 |
| Sodium (mmol/L), median (IQR) | 138.6 (136.6,141.9) | 138.5 (136.9,140.1) | 0.349 |
| Potassium (mmol/L), mean±SD | 3.81±0.57 | 3.80±0.40 | 0.910 |
| Chloride (mmol/L), median (IQR) | 105.2 (102.5,109.3) | 104.8 (102.5,106.9) | 0.420 |
| $CO_2$ (mmol/L), median (IQR) | 23.5 (18.8,25.1) | 24.6 (22.8,26.3) | 0.017 |
| Anion gap, median (IQR) | 14.4 (12.7,16.0) | 12.9 (11.7,14.4) | 0.011 |
| Cholesterol (mg/dL), mean±SD | 150±38 | 184±53 | 0.001 |
| Triglyceride (mg/dL), median (IQR) | 105 (74,184) | 104 (79,144) | 0.897 |
| HDL (mg/dL), mean±SD | 45±12 | 48±12 | 0.199 |
| LDL-c (mg/dL), mean±SD | 83±32 | 112±42 | <0.001 |

Abbreviations; EAG: Estimated average glucose, eGFR: Estimated glomerular filtration rate, $CO_2$: Carbondioxide, HDL: High-density lipoprotein cholesterol, LDL-c: Calculated low-density lipoprotein cholesterol, IHM: In-hospital mortality, IQR: Interquartile range

(range: 115–189 mg/dL), both higher than those in survived cases. Conversely, the albumin levels were 3.7 g/dL (range: 3.3–4.1 g/dL), which were lower than those in survived cases.

Besides these routine laboratory parameters, previous evidence has indicated that the novel glucose-related blood biomarkers, including the stress hyperglycemia ratio (SHR), glucose to estimated average glucose ratio (GAGR), and recently described prognostic markers such as the glucose to albumin ratio (GAR) and glucose to potassium ratio (GPR), have been increasingly utilized for the prognosis of various conditions, including cardiovascular disease, cancer, metabolic syndrome, and acute ischemic stroke [20–22]. We therefore analyzed these values for each patient. The median values of almost all novel glucose-related blood parameters, including SHR, GAGR, GAR, GPR, GUFR and TyG, in patients who died in the hospital were significantly higher than those who survived (Table 3), indicating the predictive potential of these glucose-related blood biomarkers for the severity and in-hospital mortality of AIS patients.

**Comparison of novel glucose-related blood biomarkers in AIS patients with different severity and clinical complications.**

Because our study revealed that the hospitalized deceased patients exhibited significantly higher levels of SHR, GAGR, GAR, GPR, GUFR, and TyG than the surviving patients, we subsequently conducted additional analysis and comparative studies on AIS patients categorized by disease severity (Fig 2) and clinical complications during current treatments including hemorrhagic transformation, atrial fibrillation, pneumonia, and sepsis (Fig 3). We found that only SHR, GAGR, GAR, and GPR showed statistically significant differences between mild and severe cases (Fig 2). These glucose-related blood parameters were also elevated in patients with hemorrhagic transformation and pneumonia (Fig 3). Interestingly, significantly

**Table 3. The novel glucose-related blood biomarkers in AIS patients classified by in-hospital mortality and survivor groups.**

| Novel glucose-related blood biomarkers | IHM (n=31) | Survivor (n=320) | *p*-value |
|---|---|---|---|
| Stress hyperglycemia ratio (SHR), median (IQR) | 24.5 (20.0,30.5) | 17.0 (15.2,19.0) | <0.001 |
| Glucose to EAG ratio (GAGR), median (IQR) | 1.12 (0.96,1.47) | 0.81 (0.73,0.91) | <0.001 |
| Glycemic gap (GG), median (IQR) | 42 (11,59) | 24.5 (14,34) | 0.103 |
| Glucose to albumin ratio (GAR), median (IQR) | 40.0 (30.9,50.2) | 25.0 (21.7,31.5) | <0.001 |
| Glucose to potassium ratio (GPR), median (IQR) | 40.05 (27.91,52.55) | 25.90 (22.63,33.29) | <0.001 |
| Glucose-BUN to eGFR ratio (GUFR), median (IQR) | 30.79 (18.23, 91.01) | 16.24 (10.95,27.22) | <0.001 |
| Triglyceride-glucose index (TyG), median (IQR) | 15553 (9420,27936) | 10306 (7542,15991) | 0.006 |

Abbreviations; EAG: Estimated average glucose, BUN: Blood urea nitrogen, IHM: In-hospital mortality, IQR: Interquartile range

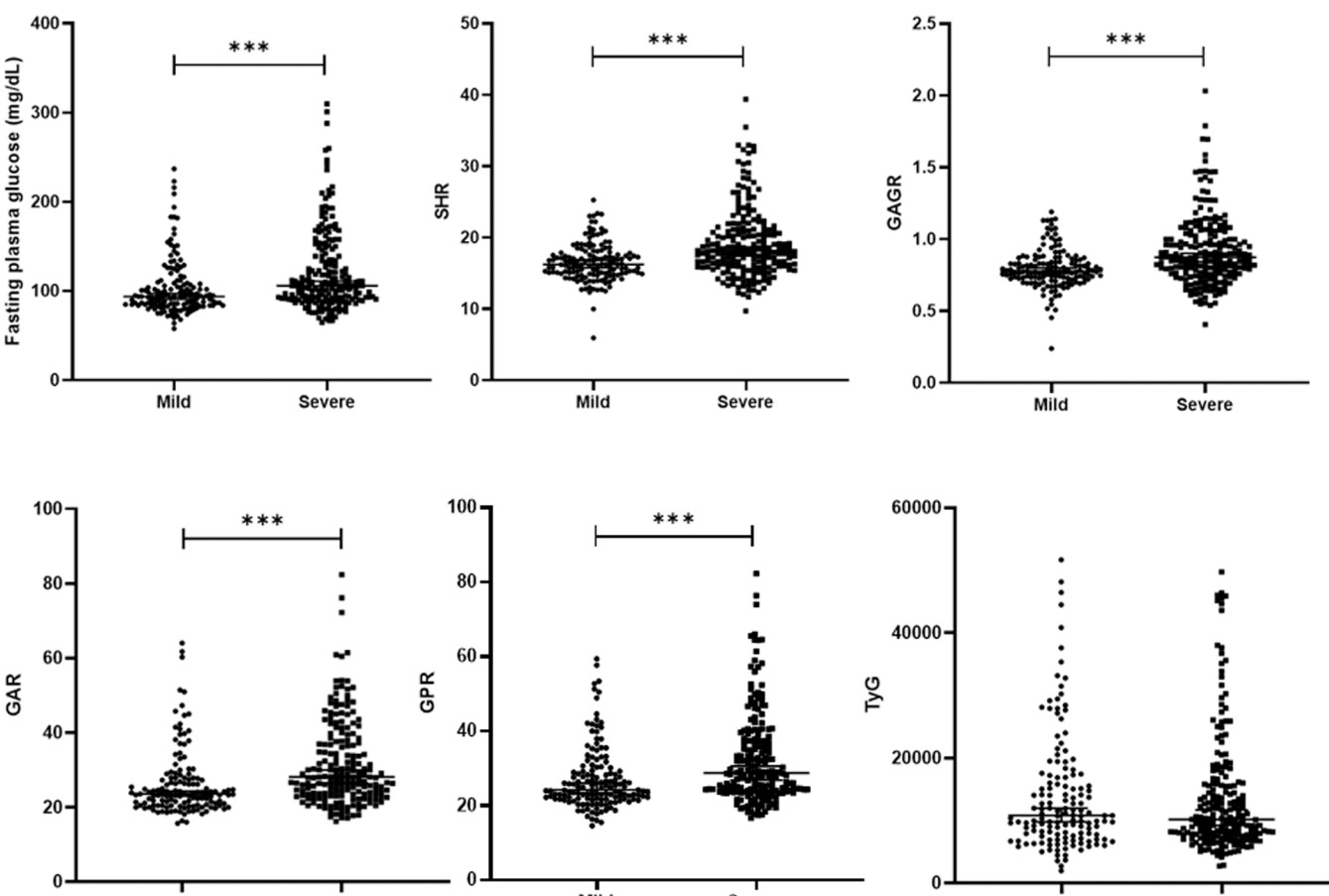

**Fig 2. Comparison of novel glucose-related blood biomarkers in patients with acute ischemic stroke categorized by mild and severe cases.**
The illustration includes fasting plasma glucose (FPG), stress hyperglycemia ratio (SHR), glucose to estimated average glucose ratio (GAGR), glucose to albumin ratio (GAR), glucose to potassium ratio (GPR), and triglyceride-glucose index (TyG). This study demonstrated that levels of FPG, SHR, GAGR, GAR, and GPR in severe cases were markedly elevated compared to mild cases. Error bars denote the median and interquartile range. Significance was analyzed using the Mann-Whitney U test, with $*p < 0.05$, $**p < 0.01$, $***p < 0.001$.

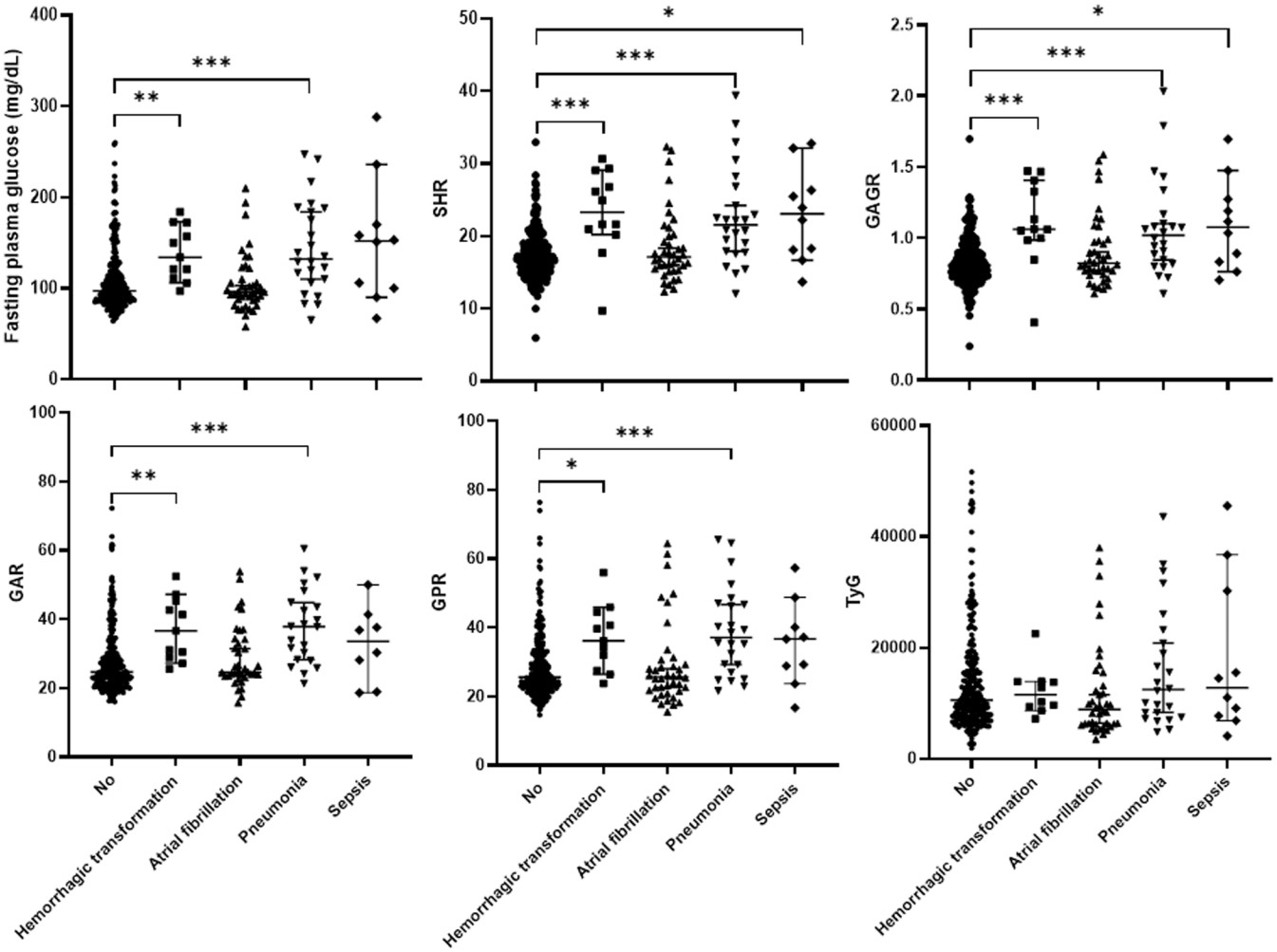

**Fig 3. Comparison of novel glucose-related blood biomarkers in acute ischemic stroke patients with different clinical complications.** This study examined the associations between the levels of FPG, SHR, GAGR, GAR, GPR, and TyG with clinical complications. Major clinical complications in AIS patients were divided into four groups: hemorrhagic transformation, atrial fibrillation, pneumonia, and sepsis. Statistical analysis was determined using the Kruskal-Wallis test, with *$p < 0.05$, **$p < 0.01$, ***$p < 0.001$, revealing statistically significant differences.

increased SHR and GAGR were observed in patients with sepsis (Fig 3). These findings may inform the selection of appropriate biomarkers for assessing severity and predicting complication in patients with acute ischemic stroke.

**Assessment of novel glucose-related blood biomarkers in predicting severity and in-hospital mortality in patients with acute ischemic stroke.**

Severe cases, defined by an NIHSS score >5 at admission baseline, exhibit various clinical manifestations, including early neurological deterioration (END) and the need for appropriate management, such as thrombolytic agents (e.g., rtPA), oxygen support, endovascular treatments, and transfer to the intensive care unit. However, delayed prediction, especially in clinical settings with limited medical resources, may contribute to in-hospital mortality. Given that SHR, GAGR, GAR, and GPR were the most significantly elevated biomarkers in these patients, we conducted a more thorough analysis of these blood biomarkers to predict the severity of patients with acute ischemic stroke. Our findings indicated that the most

reliable indicator was the glucose to albumin ratio (GAR), which had an AUC of 0.672 (95% CI: 0.614–0.731), and a cutoff value of ≥ 24.4, demonstrating a sensitivity of 72.8% and a specificity of 56.6%, in assessing patient severity. Another reliable glucose-related blood biomarker was the stress hyperglycemia ratio (SHR), which had an AUC of 0.666 (95% CI: 0.608–0.724), with a cutoff value of ≥ 16.9, indicating sensitivity and specificity for severity at 64.6% and 59.3%, respectively (Table 4 and Fig 4).

Nevertheless, SHR demonstrated the highest reliability when these parameters were employed to evaluate the in-hospital mortality of patients with acute ischemic stroke. Its AUC was 0.832 (95% CI: 0.734–0.930), and its cutoff value was ≥ 18.0, indicating a sensitivity and specificity of 87.1% and 64.7%, respectively, for predicting in-hospital mortality. The performance of the novel marker, GAR, was second-best, with an AUC of 0.825 (95% CI: 0.763–0.887), and a cutoff value of ≥ 30.0. This indicated a sensitivity and specificity of 82.8% and 72.7% for predicting in-hospital mortality, respectively. AUC values were obtained for both parameters at a very good confidence level (0.8 ≤ AUC < 0.9; Considerable) [23] (Table 4 and Fig 4).

**Assessment of the predictive value of novel glucose-related blood biomarkers in the diagnosis of stroke-associated pneumonia (SAP) and the necessity of endovascular thrombectomy (EVT) in patients with acute ischemic stroke.**

Patients with acute ischemic stroke displaying severe symptoms may encounter significant clinical consequences, including stroke-associated pneumonia (SAP), as well as limitations and risks associated with surgical interventions such as endovascular thrombectomy during treatment. Timely and effective assessment of these causes and conditions may mitigate the severity and mortality rates of patients in the hospital. Employing novel glucose-related blood biomarkers to evaluate the risk of stroke-associated pneumonia and to assess the need of endovascular thrombectomy may provide additional benefits for patients. We further conducted an analysis of these markers to assess the occurrence of stroke-associated pneumonia and the requirement for endovascular thrombectomy. The novel glucose-related blood biomarkers exhibiting the highest AUC were the SHR, with an AUC of 0.817 (95% CI: 0.737–0.869). A cutoff value of ≥ 18.0 demonstrated a high sensitivity of 82.5% and a specificity of 65.6% for predicting stroke-associated pneumonia (SAP). Moreover,

**Table 4. The evaluation of severity and in-hospital mortality in patients with acute ischemic stroke using novel glucose-related blood biomarkers.**

| Variables | AUC (95% CI) | Cutoff | %Sens | %Spec | %NPV | %PPV | Accuracy | p-value |
|---|---|---|---|---|---|---|---|---|
| Severity of AIS (n = 191) | | | | | | | | |
| FPG | 0.640 (0.579,0.700) | 98 | 63.1 | 58.6 | 52.8 | 68.4 | 61.3 | <0.001 |
| SHR | 0.666 (0.608,0.724) | 16.9 | 64.6 | 59.3 | 54.1 | 69.3 | 62.4 | <0.001 |
| GAGR | 0.659 (0.601,0.717) | 0.80 | 68.4 | 57.2 | 56.1 | 69.5 | 63.8 | <0.001 |
| GAR | 0.672 (0.614,0.731) | 24.4 | 72.8 | 56.6 | 59.4 | 70.4 | 66.1 | <0.001 |
| GPR | 0.648 (0.588,0.707) | 26.00 | 60.0 | 57.9 | 50.6 | 66.8 | 59.1 | <0.001 |
| In-hospital mortality (n = 31) | | | | | | | | |
| FPG | 0.790 (0.707,0.873) | 110 | 80.6 | 68.4 | 97.3 | 19.8 | 69.5 | <0.001 |
| SHR | 0.832 (0.734,0.930) | 18.0 | 87.1 | 64.7 | 98.1 | 19.3 | 66.7 | <0.001 |
| GAGR | 0.822 (0.722,0.922) | 0.90 | 80.6 | 71.2 | 97.4 | 21.4 | 72.1 | <0.001 |
| GAR | 0.825 (0.763,0.887) | 30.0 | 82.8 | 72.7 | 97.8 | 22.6 | 73.6 | <0.001 |
| GPR | 0.758 (0.678,0.837) | 28.70 | 74.2 | 62.2 | 96.1 | 16.0 | 63.1 | <0.001 |

Abbreviations; FPG: Fasting plasma glucose, SHR: Stress hyperglycemia ratio, GAGR: Glucose to estimated average glucose, GAR: Glucose to albumin ratio, GPR: Glucose to potassium ratio, AIS: acute ischemic stroke, AUC: Area under curve, Sens: Sensitivity, Spec: Specificity, NPV: Negative predictive value, PPV: Positive predictive value, 95% CI: 95% confidential interval.

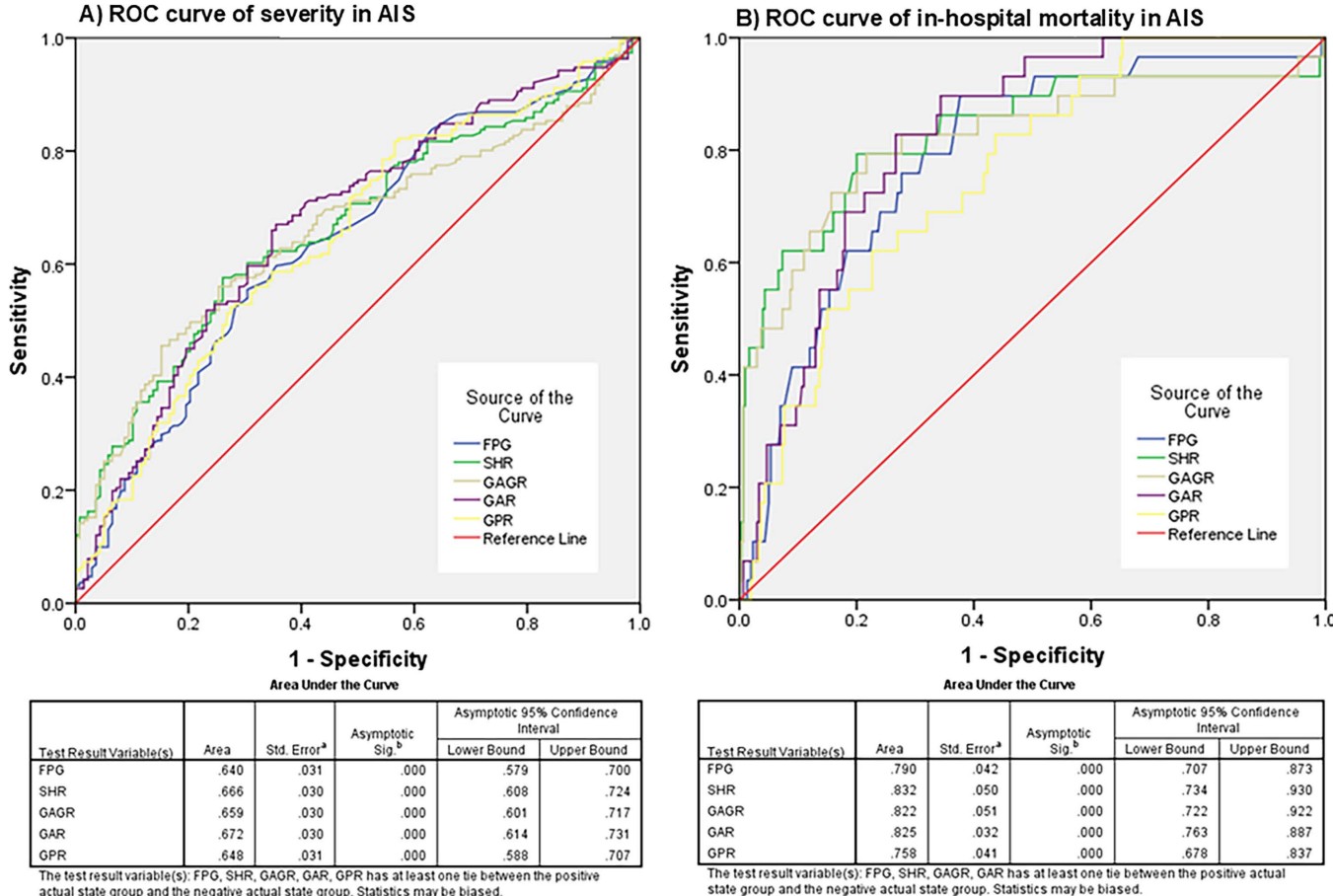

Fig 4. Receiver operating characteristic (ROC) curve analysis of novel glucose-related blood biomarkers for the prediction of severity and in-hospital mortality in patients with acute ischemic stroke (AIS). The results indicated that the area under the curve (AUC) values for fasting plasma glucose (FPG), stress hyperglycemia ratio (SHR), glucose to estimated average glucose ratio (GAGR), glucose to albumin ratio (GAR), and glucose to potassium ratio (GPR) in predicting severity were 0.640, 0.666, 0.659, 0.672, and 0.648, respectively. The ROC analysis showed that GAR had the highest AUC value for predicting severity in AIS patients (A). For predicting in-hospital mortality, the AUC values for FPG, SHR, GAGR, GAR, and GPR were 0.790, 0.832, 0.822, 0.825, and 0.758, respectively, with SHR demonstrating the greatest AUC for predicting in-hospital mortality among AIS patients (B).

the novel marker glucose to albumin ratio (GAR) exhibited an AUC of 0.798 (95% CI: 0.719–0.881), with a cutoff value of ≥ 30.0 indicating a high sensitivity of 78.9% and specificity of 73.9% (Table 5 and Fig 5). SHR and GAR may be useful for informing the management of oxygen support therapy such as mechanical ventilation. Conversely, the most reliable indicator for determining the necessity of endovascular thrombectomy was glucose to estimated average glucose (GAGR), which exhibited an AUC of 0.735 (95% CI: 0.642–0.829) and a cutoff value of ≥ 0.85, indicating a sensitivity of 76.5% and specificity of 59.6%. In comparison, the SHR exhibited an AUC of 0.713 (95% CI: 0.617–0.829) with a cutoff value of ≥ 17.5, showing a sensitivity of 67.6% and specificity of 56.5%. For endovascular therapy, clinical and radiological examinations are the primary criteria for decision-making in the appropriate management. However, this result may provide valuable data on laboratory markers as supportive tools for clinical management, especially in settings with limited resources (Table 5 and Fig 5).

**Table 5. The evaluation of novel glucose-related blood biomarkers for predicting stroke-associated pneumonia (SAP) and the requirement of endovascular thrombectomy (EVT) in patients with acute ischemic stroke.**

| Variables | AUC (95% CI) | Cutoff | %Sens | %Spec | %NPV | %PPV | Accuracy | *p*-value |
|---|---|---|---|---|---|---|---|---|
| AIS associated pneumonia (n = 40) | | | | | | | | |
| FPG | 0.757 (0.673,0.841) | 110 | 75.0 | 69.1 | 95.6 | 23.8 | 69.80 | <0.001 |
| SHR | 0.817 (0.737,0.869) | 18.0 | 82.5 | 65.6 | 96.7 | 23.6 | 67.5 | <0.001 |
| GAGR | 0.800 (0.719,0.881) | 0.90 | 75.0 | 72.0 | 95.7 | 25.6 | 72.4 | <0.001 |
| GAR | 0.798 (0.731,0.865) | 30.0 | 78.9 | 73.9 | 96.4 | 28.3 | 74.5 | <0.001 |
| GPR | 0.764 (0.691,0.837) | 28.70 | 72.5 | 62.9 | 94.7 | 20.1 | 64.0 | <0.001 |
| Treated with EVT (n = 34) | | | | | | | | |
| FPG | 0.623 (0.529,0.717) | 100 | 64.7 | 50 | 92.9 | 12.1 | 51.0 | 0.027 |
| SHR | 0.713 (0.617,0.809) | 17.5 | 67.6 | 56.5 | 94.2 | 14.3 | 57.5 | <0.001 |
| GAGR | 0.735 (0.642,0.829) | 0.85 | 76.5 | 59.3 | 95.9 | 16.8 | 61.0 | <0.001 |
| GAR | 0.663 (0.573,0.754) | 27.8 | 70.0 | 61.2 | 95.3 | 15.3 | 62.0 | 0.003 |
| GPR | 0.652 (0.556,0.748) | 28.85 | 61.8 | 61.7 | 93.8 | 14.8 | 61.7 | 0.006 |

Abbreviations; FPG: Fasting plasma glucose, SHR: Stress hyperglycemia ratio, GAGR: Glucose to estimated average glucose, GAR: Glucose to albumin ratio, GPR: Glucose to potassium ratio, AIS: Acute ischemic stroke, EVT: Endovascular thrombectomy, AUC: Area under curve, Sens: Sensitivity, Spec: Specificity, NPV: Negative predictive value, PPV: Positive predictive value, 95% CI: 95% confidence interval.

**Univariate and multivariate analyses of novel glucose-related blood biomarkers with in-hospital mortality risk of AIS patients.**

Based on our findings, we utilized both univariate and multivariate analyses to assess novel glucose-related blood biomarkers with other risk factors, presenting the outcomes as odds ratios (OR) and adjusted odds ratios (AOR). The analysis includes three distinct models: Model A consisted of variables such as age, sex, systolic blood pressure, diastolic blood pressure, and length of hospital stay; Model B comprised age, NIHSS score, and mRS score; while Model C incorporated age, hypertension, and random blood glucose levels. We found that AIS patients with a SHR ≥ 18.0 exhibited a markedly greater likelihood of in-hospital mortality, with an odds ratio of 12.365 compared to those with a SHR < 18.0 (OR = 12.365, 95% CI: 4.221–36.222, p < 0.001). The adjusted odds of in-hospital mortality, when combined with the covariates in Models A, B, and C, were 9.698, 6.804, and 11.582 times greater, respectively. Furthermore, AIS patients with a GAR ≥ 30.0 exhibited a 12.761-fold increased risk of in-hospital mortality (OR = 12.761, 95% CI: 4.711–34.563, p < 0.001). After adjustment for the variables in Models A, B, and C, the AOR for in-hospital mortality were 7.945, 7.151, and 11.127, respectively. These data indicated that novel glucose-related blood biomarkers, in conjunction with various risk factors, substantially elevate the likelihood of in-hospital mortality (Table 6).

## Discussion

Acute ischemic stroke (AIS) is not only the second leading cause of death worldwide but also the primary source of acquired disability in the adult population [24–26]. In recent years, blood biomarkers have been identified, and numerous promising candidates have been proposed as predictors of stroke [27,28]. However, no individual blood biomarker has yet demonstrated sufficient sensitivity and specificity for routine clinical use. This study investigated the clinical utility of novel glucose-related blood biomarkers in predicting the severity and in-hospital mortality of patients with AIS. The findings highlighted significant associations between these biomarkers and patient outcomes, demonstrating their potential value in clinical practice.

To our knowledge, few studies have examined the association between glucose-related blood biomarkers and prognosis in patients with AIS. In this study, we identified SHR as the most reliable biomarker for predicting in-hospital mortality

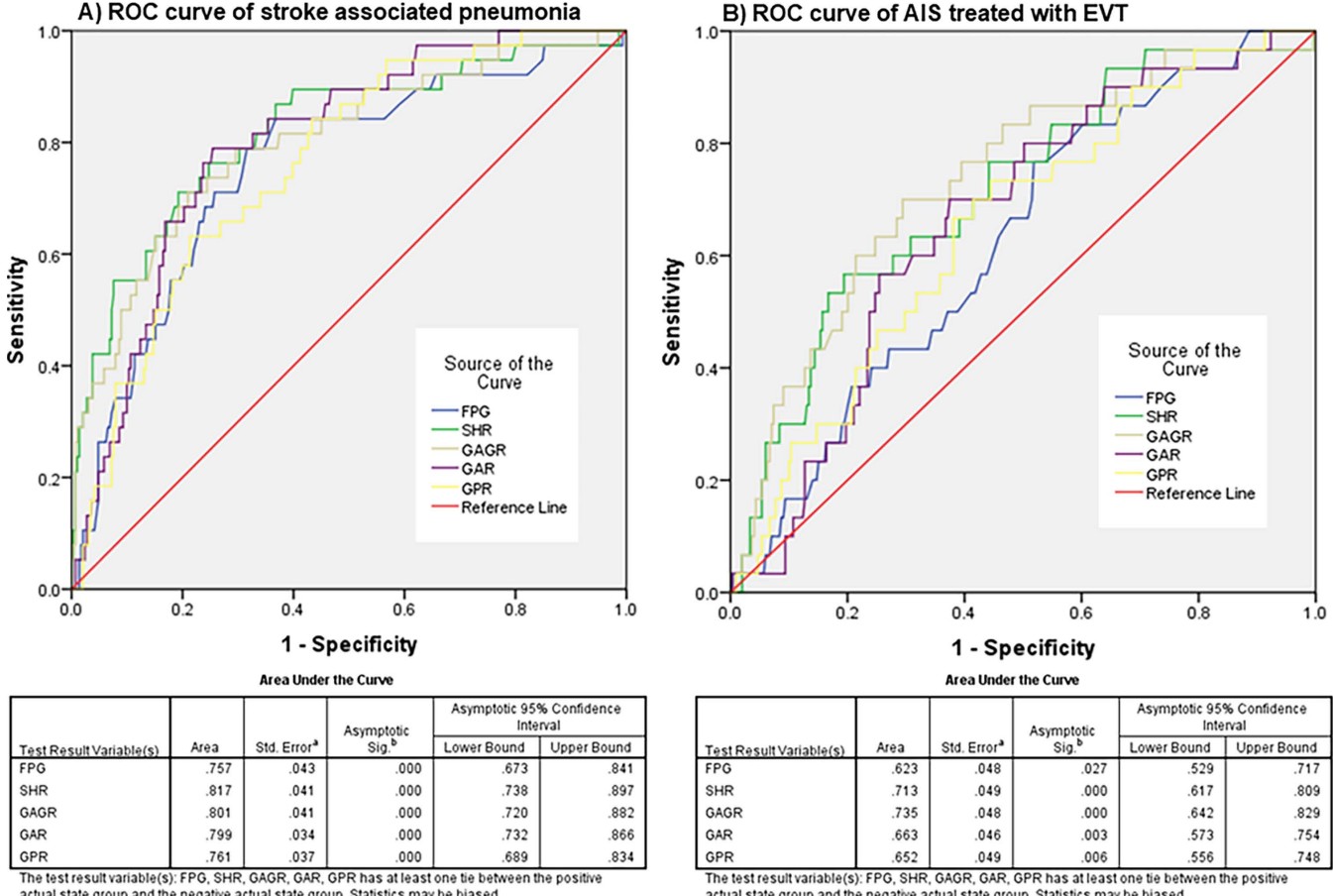

**Fig 5. Receiver operating characteristic (ROC) curve analysis for evaluation of novel glucose-related blood biomarkers for predicting clinical complications, specifically stroke-associated pneumonia (SAP) and the need for therapeutic management with endovascular thrombectomy (EVT).** To assess the predictive ability of these markers, including fasting plasma glucose (FPG), stress hyperglycemia ratio (SHR), glucose to estimated average glucose ratio (GAGR), glucose to albumin ratio (GAR), and glucose to potassium ratio (GPR), the AUC values for SAP were 0.757, 0.817, 0.800, 0.798, and 0.764, respectively. The SHR exhibited the highest AUC value for screening SAP (A). In patients with acute ischemic stroke (AIS) treated with EVT, the ROC analysis demonstrated AUC values of 0.623, 0.713, 0.735, 0.663, and 0.652 for FPG, SHR, GAGR, GAR, and GPR, respectively. Notably, the best AUC value for predicting the requirement for EVT was found for GAGR (B).

(IHM), with an area under the curve (AUC) of 0.832 and a cutoff value of ≥18.0, yielding sensitivity and specificity rates of 87.1% and 64.7%, respectively. SHR reflects the balance between fasting plasma glucose and hemoglobin A1c, providing a stable metric for evaluating acute glycemic fluctuations during stress [29]. Elevated SHR values indicate a heightened neurohormonal response and systemic inflammation [9,30]. A recent systematic review and meta-analysis demonstrated the relationship between SHR and poor neurological outcomes as well as mortality in AIS patients [12]. A first study by Roberts et al. found that an admission SHR ≥ 1.14 was indicative of poor outcomes in hospitalized patients [31]. Furthermore, SHR has been associated with increased 30-day and 90-day mortality in AIS patients, underscoring its potential value in risk stratification [32]. In Thailand, Krongsut S. et al. showed that elevated SHR was significantly associated with increased risks of in-hospital mortality, malignant cerebral edema, symptomatic intracerebral hemorrhage, 3-month mortality, and poor functional outcomes in thrombolyzed acute ischemic stroke patients, and SHR exhibited superior predictive performance for these fatal outcomes compared to traditional glucose metrics, including fasting plasma glucose,

**Table 6. The risk ratio of in-hospital mortality in AIS patients with higher cut-off value of novel glucose-related blood biomarkers.**

| Markers of in-hospital mortality | Univariate | | Multivariate | | | | | |
|---|---|---|---|---|---|---|---|---|
| | | | Model A | | Model B | | Model C | |
| | OR (95% CI) | p-value | AOR (95% CI) | p-value | AOR (95% CI) | p-value | AOR (95% CI) | p-value |
| FPG ≥ 110 | 9.035 (3.594,22.708) | <0.001 | 7.316 (2.774,19.298) | <0.001 | 5.366 (1.936,14.872) | 0.001 | 8.637 (3.233,23.074) | <0.001 |
| SHR ≥ 18.0 | 12.365 (4.221,36.222) | <0.001 | 9.698 (3.175,29.617) | <0.001 | 6.804 (2.140,21.628) | 0.001 | 11.582 (3.887,34.512) | <0.001 |
| GAGR ≥ 0.90 | 10.326 (4.102,25.997) | <0.001 | 8.003 (3.047,21.017) | <0.001 | 4.802 (1.744,13.224) | 0.002 | 9.137 (3.569,23.391) | <0.001 |
| GAR ≥ 30.0 | 12.761 (4.711,34.563) | <0.001 | 7.945 (2.807,22.484) | <0.001 | 7.151 (2.401,21.295) | <0.001 | 11.127 (3.916,31.618) | <0.001 |
| GPR ≥ 28.7 | 4.705 (2.040,10.850) | 0.001 | 3.367 (1.394,8.129) | 0.007 | 3.603 (1.347,9.641) | 0.011 | 3.886 (1.639,9.215) | 0.002 |

Abbreviations: FPG, Fasting plasma glucose; SHR, Stress hyperglycemia ratio; GAGR, Glucose to estimated average glucose; GAR, Glucose to albumin ratio; GPR, Glucose to potassium ratio; OR, odds ratio; AOR, Adjusted odds ratio.

admission random plasma glucose, and glycated hemoglobin (HbA1c) [19]. In conjunction with our findings, SHR may serve as a key predictor of severity and mortality in AIS patients.

Interestingly, a novel biomarker, GAR, was demonstrated to be highly effective in assessing both severity and mortality in AIS patients. GAR exhibited an AUC of 0.672 with a cutoff value of 24.4, predicting severity with sensitivity and specificity rates of 72.8% and 56.6%, respectively. Additionally, GAR demonstrated an AUC of 0.825 with a cutoff value of 30.0, forecasting in-hospital mortality with sensitivity and specificity rates of 82.8% and 72.7%, respectively. A related study revealed that GAR outperformed other biomarkers in predicting 90-day mortality in intracranial hemorrhagic stroke (ICH) patients (AUC = 0.72) [14]. GAR integrates fasting glucose and albumin levels, with the latter serving as a biomarker of nutritional status and systemic inflammation [21,33]. A recent study investigated the admission blood glucose-to-albumin ratio (AAR) as a predictor of futile recanalization in patients with acute cerebral infarction due to anterior circulation large vessel occlusion who underwent successful interventional recanalization [34]. The results demonstrated that higher AAR tertiles were significantly associated with increased rates of futile recanalization, defined as a modified Rankin Scale (mRS) score of 3–6 at three months post-procedure [34]. Moreover, combining AAR tertiles with admission NIHSS scores and collateral circulation status resulted in a predictive model with high accuracy for futile recanalization (AUC = 0.907) [34]. However, hypoalbuminemia, commonly observed in cardiovascular events such as ischemic or hemorrhagic stroke, correlated with compromised vascular integrity and a pro-inflammatory state, exacerbating the detrimental effects of hyperglycemia [35]. The strong association between GAR and patient outcomes underscores its dual role as both a metabolic and inflammatory biomarker. These findings corroborate and expand upon previous research regarding glycemic markers in cerebrovascular diseases. While SHR and GAR outperformed other biomarkers such as the glucose-to-potassium ratio (GPR) and glucose-to-estimated average glucose ratio (GAGR), further investigation in larger and more diverse cohorts is warranted.

SHR and GAR are biomarkers used to assess stress hyperglycemia and its impact on acute ischemic stroke (AIS) [19,34]. However, they differ in their calculation, physiological basis, and clinical significance. SHR represents stress-induced hyperglycemia by adjusting for pre-existing glycemic control (HbA1c) and has more dynamic change of glucose level [12,31] while GAR integrates both hyperglycemia and hypoalbuminemia [21,33], indicating both metabolic and inflammatory stress for AIS patients [34]. Incorporating SHR and GAR into routine clinical assessments offers several practical advantages. Both biomarkers rely on commonly available biochemical parameters, making them accessible and cost-effective tools for early risk

stratification, particularly in resource-limited settings of AIS patients. By identifying high-risk patients upon admission, these markers can guide timely therapeutic decisions, including intensified monitoring, targeted glycemic control, and advanced interventions such as endovascular thrombectomy (EVT) [10,36,37]. Additionally, they can aid in anticipating severe complications such as stroke-associated pneumonia [38]. SHR and GAR, with cutoff values of ≥17.5 and ≥27.8, respectively, may indicate the need for EVT. Furthermore, SHR ≥ 18.0 and GAR ≥ 30.0 were the most effective in predicting disease deterioration associated with stroke-associated pneumonia. Thus, SHR and GAR could provide complementary insights when used alongside established clinical scales, such as the National Institutes of Health Stroke Scale (NIHSS) and the modified Rankin Scale (mRS) [39]. Their predictive accuracy for severity and mortality highlights their potential to enhance existing prognostic models, ultimately improving functional outcomes for AIS patients.

Despite these strengths, certain limitations must be acknowledged. The retrospective design and single-center nature of this study may limit the generalizability of these findings. Selection bias cannot be entirely excluded, and the relatively small sample size may have affected the robustness of the analyses. The cross-sectional measurement of biomarkers precludes an understanding of their dynamic changes over time, which could provide additional prognostic value. This study did not include data on clinical characteristics for mild and severe patients, nor on ischemic time. Moreover, we did not perform a relationship between NIHSS and these biomarkers and conduct comprehensive statistical analyses, such as multivariate analysis for ROC curves, p-value adjustments for multiple variables, or the provision of confidence intervals for sensitivity and specificity. Furthermore, diabetes, a strong confounder, was not incorporated into our multivariate analysis. Future research should focus on validating these biomarkers in larger, multicenter cohorts, exploring their temporal trends during both the acute and recovery phases of AIS, and including comprehensive statistical analysis. Investigating the interplay between glycemic markers and other systemic factors, such as inflammation and oxidative stress, could further elucidate their roles in stroke pathophysiology. Additionally, randomized controlled trials evaluating targeted interventions based on SHR and GAR levels could provide evidence supporting their clinical utility in personalized treatment strategies.

## Conclusion

This study highlights the potential prognostic value of SHR and GAR in patients with AIS as these parameters demonstrated an association with in-hospital mortality. In resource-limited settings, SHR and GAR may be utilized to systematically record patient information and help gather critical data on stroke incidence and complication. Their integration into routine clinical practice could enhance early risk stratification, optimize resource allocation, and improve clinical outcomes. However, further validation through large-scale, multivariate analyses, and prospective studies is warranted to confirm these findings and refine their application in diverse clinical settings.

## Supporting information

**S1 File. The raw data of this study.**
(XLSX)

## Acknowledgments

We deeply thank the Department of Medical Technology and Clinical Pathology of Saraburi Hospital for their support and appreciate the Department of Medical Technology, Faculty of Allied Health Sciences, Thammasat University for their strong collaboration of this research.

## Author contributions

**Conceptualization:** Peerapong Kamjai, Pornpimon Angkasekwinai.

**Data curation:** Peerapong Kamjai.

**Formal analysis:** Peerapong Kamjai, Pornpimon Angkasekwinai.

**Funding acquisition:** Pornpimon Angkasekwinai.

**Investigation:** Peerapong Kamjai, Pornpimon Angkasekwinai.

**Methodology:** Peerapong Kamjai, Pornpimon Angkasekwinai.

**Project administration:** Peerapong Kamjai, Pornpimon Angkasekwinai.

**Supervision:** Pornpimon Angkasekwinai.

**Validation:** Peerapong Kamjai, Pornpimon Angkasekwinai.

**Visualization:** Peerapong Kamjai, Pornpimon Angkasekwinai.

**Writing – original draft:** Peerapong Kamjai, Pornpimon Angkasekwinai.

**Writing – review & editing:** Peerapong Kamjai, Pornpimon Angkasekwinai.

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
