## [Decision Letter · Decision Letter 0]

9 Feb 2025

PONE-D-25-01318Clinical utility of novel glucose-related blood biomarkers for predicting in-hospital mortality of patients with acute ischemic strokePLOS ONE

Dear Dr. Angkasekwinai,

Thank you for submitting your manuscript to PLOS ONE. After careful consideration, we feel that it has merit but does not fully meet PLOS ONE’s publication criteria as it currently stands. Therefore, we invite you to submit a revised version of the manuscript that addresses the points raised during the review process.

We look forward to receiving your revised manuscript.

Kind regards,

Atakan Orscelik

Academic Editor

PLOS ONE

2. We note that there is identifying data in the Supporting Information file <S1_File.xlsx>. Due to the inclusion of these potentially identifying data, we have removed this file from your file inventory. Prior to sharing human research participant data, authors should consult with an ethics committee to ensure data are shared in accordance with participant consent and all applicable local laws.

-Location data

Please remove or anonymize all personal, ensure that the data shared are in accordance with participant consent, and re-upload a fully anonymized data set. Please note that spreadsheet columns with personal information must be removed and not hidden as all hidden columns will appear in the published file.

Reviewers' comments:

Reviewer's Responses to Questions

**Comments to the Author**

1. Is the manuscript technically sound, and do the data support the conclusions?

Reviewer #1: Yes

Reviewer #2: Partly

2. Has the statistical analysis been performed appropriately and rigorously? 

Reviewer #1: Yes

Reviewer #2: No

3. Have the authors made all data underlying the findings in their manuscript fully available?

Reviewer #1: Yes

Reviewer #2: Yes

4. Is the manuscript presented in an intelligible fashion and written in standard English?

Reviewer #1: Yes

Reviewer #2: Yes

5. Review Comments to the Author

Reviewer #1: Dear Author

The manuscript's language is clear but could benefit from minor edits for grammatical consistency. For example: Replace "remain minimized information in Thailand" with "has limited information in Thailand." Change "guiding in-hospital mortality" to "in predicting in-hospital mortality." Include a ROC curve figure to visually represent the AUC of SHR and GAR. Add a flowchart summarizing patient selection and analysis to improve clarity.

1. The introduction provides a good rationale for the study but could benefit from a clearer linkage between hyperglycemia, glycemic markers, and AIS outcomes. Consider elaborating on why current markers (e.g., SHR) are insufficient and why novel glucose-related markers are needed in clinical practice. The significance of the study in the context of Thailand is mentioned but could be expanded to provide more context on the prevalence of AIS and hyperglycemia in the region.

2. Clearly define how "severe cases" of AIS were classified. Did you use the National Institutes of Health Stroke Scale (NIHSS) or another severity index? Describe how the cutoff values for GAR and SHR were determined. Was this based on a receiver operating characteristic (ROC) curve analysis? Provide more detail about the inclusion/exclusion criteria for the retrospective analysis to ensure reproducibility. Indicate whether ethical approval was obtained for the retrospective study and how patient confidentiality was maintained.

3. While the AUC values and cutoff points for SHR and GAR are reported, the manuscript should include a comparative analysis of the two biomarkers (e.g., their strengths and limitations in clinical prediction). Provide confidence intervals for sensitivity and specificity values to enhance statistical rigor. Consider including a table summarizing patient characteristics (e.g., age, sex, comorbidities) for better understanding of the study cohort.

4. The discussion should emphasize the clinical utility of GAR and SHR and their potential integration into existing triage protocols. Compare your findings to those of other studies to highlight similarities or differences and their implications. Discuss limitations, such as the retrospective design, single-center study, and potential confounding factors.

5. The conclusion is strong but could explicitly recommend how GAR and SHR could be applied in resource-limited settings to improve AIS management.

Reviewer #2: This is an interesting study by the authors covering the potential of glucose based prediction of stroke associated endpoints. It is a single-center retrospective analysis from Thailand. However, I have several major points.

Abstract

Mention that it is a single center study

Provide examples of novel glucose biomarkers by name

Why are different cut-offs used to present AUC and OR for albumin to glucose

Ratio – please correct.

Why is AGR considered superior if the AUC is lower than SHR?

-this needs rephrasing

I think the presented data do not merit the word “predictive” – call it “associated”. It is a retrospective study with lots of bias

Introduction:

Line 61: do you mean fatal outcomes?

Line 64-66: far too bold statement. Data provided may at best inform on prognosis but will not influence guidelines and I highly doubt it will change management particularly the acute setting. Rephrase.

Methods:

Which glucose measurement at which time point exactly was used for analysis? Was it always the same time window?

What were the criteria of determining stroke severity? Which NIHSS Scores etc.? The current phrasing is subject to strong interpretation.

Was there any stratification for baseline characteristics performed? What about multivariate regression? This seems to have been carried out (Table 6) but it isn’t described.

Overal the statistics section needs to explain every statistical model and method that was employed!

What was the primary outcome? Death or severity? What were the power calculations?

Was there adjustment of p-value for multiple testing? The authors are basically testing several variables against a several outcomes

Results:

Overall, the number of figures and tables is too much and some of the data is redundant. This needs to be re-designed to avoid redundancies.

Table 1 – Include ischemic time if available. This will be one of the main drivers of mortality

Table1 – only shows survivors vs. non-survivors. A table showing clinical characteristics of the so called severe and mild groups is also required

What were the main causes of death?

Line 152 – how can glucose ratios be used for the diagnosis of cerebrovascular disease? Rephrase.

Figure 1: why is survived vs. dead included here for GAR although it is already shown in Table 3?

Figure 2. How does this figure help in interpreting the data? There are already 2 significant differences in FPG. SHR and GAGR only show 3 sign. differences. what is the point of showing this?

Line 205-213: I cannot interpret this, as there is no definition of stroke severity. You could analyze correlation with NIHSS severity or some other objective parameter (e.g., extent of infarcted area on imaging). Furthermore, the AUC is very low.

Table 4: is this simple logistic regression?

Why are no ROC curves provided for multivariate analysis? This is the more interesting?

table 5: why would you use glucose parameters to predict the necessity of endovascular therapy? The decision is usually based on clinical and radiological parameters.

Table 6. Why does none of the models contain the diagnosis of diabetes and obesity, which will most likely be strong confounders?

Why are no further parameters of the multivariate logistic regression shown, e.g. ROC curves for the more promising quotients?

Moreover, GAR does not do so well in multivariate model A, so I am not so sure about an improved prognostic value compared to the other parameters.

Conclusion

The first sentence is far too bold given the small sample size, inherent bias etc.

Replace the first sentence, for example, with the following: “This study highlights the potential prognostic value of SHR and GAR in patients with AIS as these parameters demonstrated an association with in-hospital mortality.” The rest of the conclusion is fine.

6. PLOS authors have the option to publish the peer review history of their article (what does this mean? ). If published, this will include your full peer review and any attached files.

**Do you want your identity to be public for this peer review?** For information about this choice, including consent withdrawal, please see our Privacy Policy .

Reviewer #1: No

Reviewer #2: No

---

## [Author Response · Author response to Decision Letter 0]

25 Mar 2025

Point-by-point Reply

Review Comments to the Author

Reviewer #1:

Dear Author

The manuscript's language is clear but could benefit from minor edits for grammatical consistency. For example: Replace "remain minimized information in Thailand" with "has limited information in Thailand." Change "guiding in-hospital mortality" to "in predicting in-hospital mortality." Include a ROC curve figure to visually represent the AUC of SHR and GAR. Add a flowchart summarizing patient selection and analysis to improve clarity.

Response: We appreciate the reviewer’s comments. We have carefully revised the current manuscript and updated the sentences according to the reviewer’s suggestions. Additionally, we have included a flowchart summarizing patient selection and a ROC curve to demonstrate the AUC of SHR and GAR, presented in Figure 1 and Figure 4, respectively, in the revised manuscript.

1. The introduction provides a good rationale for the study but could benefit from a clearer linkage between hyperglycemia, glycemic markers, and AIS outcomes. Consider elaborating on why current markers (e.g., SHR) are insufficient and why novel glucose-related markers are needed in clinical practice. The significance of the study in the context of Thailand is mentioned but could be expanded to provide more context on the prevalence of AIS and hyperglycemia in the region.

Response: We thank the reviewer for these suggestions. We have revised the introduction with a clearer context in the current manuscript as suggested by reviewer in line 56-69.

2. Clearly define how "severe cases" of AIS were classified. Did you use the National Institutes of Health Stroke Scale (NIHSS) or another severity index? Describe how the cutoff values for GAR and SHR were determined. Was this based on a receiver operating characteristic (ROC) curve analysis? Provide more detail about the inclusion/exclusion criteria for the retrospective analysis to ensure reproducibility. Indicate whether ethical approval was obtained for the retrospective study and how patient confidentiality was maintained.

Response: We sincerely appreciate the reviewer’s comments. We have used the NIHSS and definitive diagnoses from neurologists to classify severe cases. This information was included in line 90-93 in the revised manuscript.

For cutoff determination, we performed an analysis based on the ROC curve, considering appropriate sensitivity and specificity. We have included more information in the M&M section in this revised manuscript in line 120-130.

Additionally, we have provided more detailed inclusion and exclusion as demonstrated in the current manuscript in line 75-84.

Furthermore, we have revised the ethical statement to ensure clear confirmation of human subject protection as demonstrated in the current manuscript line 106-107.

3. While the AUC values and cutoff points for SHR and GAR are reported, the manuscript should include a comparative analysis of the two biomarkers (e.g., their strengths and limitations in clinical prediction). Provide confidence intervals for sensitivity and specificity values to enhance statistical rigor. Consider including a table summarizing patient characteristics (e.g., age, sex, comorbidities) for better understanding of the study cohort.

Response: We thank the reviewer for taking the time to review our work. We have focused on the AUC with a 95% CI to assess the ability of the biomarkers. We acknowledge this limitation and address confidential intervals for sensitivity and specificity in future studies (line 410-415). Regarding the summary of characteristic data, we have included patient characteristics in Table 1 in this revised manuscript.

4. The discussion should emphasize the clinical utility of GAR and SHR and their potential integration into existing triage protocols. Compare your findings to those of other studies to highlight similarities or differences and their implications. Discuss limitations, such as the retrospective design, single-center study, and potential confounding factors.

Response: We sincerely appreciate the reviewer’s comments. We include more discussion on the GAR and SHR in the discussion in line 352-359, 368-376, 384-390 and 410-415 in the revised manuscript.

5. The conclusion is strong but could explicitly recommend how GAR and SHR could be applied in resource-limited settings to improve AIS management.

Response: We appreciate the reviewer's comments. We have revised the conclusion accordingly to fit our context.

Reviewer #2:

This is an interesting study by the authors covering the potential of glucose-based prediction of stroke associated endpoints. It is a single-center retrospective analysis from Thailand. However, I have several major points.

Response: We sincerely appreciate the reviewer’s comments. We have carefully revised the current manuscript and rephrased the sentences according to reviewer’s suggestions as below.

Abstract

Mention that it is a single center study

Response: We have revised the abstract following reviewer’s comment. (Line 27-28)

Provide examples of novel glucose biomarkers by name

Response: We have revised the abstract following reviewer’s suggestion. (Line 24-25)

Why are different cut-offs used to present AUC and OR for albumin to glucose Ratio – please correct.

Response: We thank the reviewer for this comment. The cutoff value was determined and chosen by Youden’s index based on the appropriate sensitivity and specificity from ROC curve analysis. Therefore, the different cutoff values for severe cases and in-hospital mortality groups were used according to the ROC analysis. This detail has been incorporated into the materials and methods section in line 120-122 in this revised manuscript.

Why is AGR considered superior if the AUC is lower than SHR?

-these needs rephrasing

Response: We appreciate the reviewer’s feedback. In the present manuscript, we have revised abstract for better clarification. Our findings show that the GAR (glucose-to-albumin ratio) has the highest AUC for severe cases, while it is inferior to SHR in predicting in-hospital mortality.

I think the presented data do not merit the word “predictive” – call it “associated”. It is a retrospective study with lots of bias

Response: We agree and have revised the current manuscript title as “Evaluation of novel glucose-related blood biomarkers for predicting in-hospital mortality in patients with acute ischemic stroke.”

Introduction:

Line 61: do you mean fatal outcomes?

Response: Thanks for correcting this word. We meant "fatal outcomes" and have revised it accordingly, as per the reviewer’s comment.

Line 64-66: far too bold statement. Data provided may at best inform on prognosis but will not influence guidelines and I highly doubt it will change management particularly the acute setting. Rephrase.

Response: The present manuscript has been revised in accordance with the reviewer’s recommendation.

Methods:

Which glucose measurement at which time point exactly was used for analysis? Was it always the same time window?

Response: We collected fasting plasma glucose data at the same time after patients were admitted to the Stroke Unit (line 75-84).

What were the criteria of determining stroke severity? Which NIHSS Scores etc.? The current phrasing is subject to strong interpretation.

Response: We appreciate the reviewer’s feedback. We determined stroke severity using NIHSS scores. Severe cases were defined as those with an NIHSS score >5 at admission baseline (moderate to severe patients) and experiencing complications such as early neurological deterioration (END), requiring oxygen support, or transfer to the intensive care unit. This information was included in the revised manuscript in line 90-93.

Was there any stratification for baseline characteristics performed? What about multivariate regression? This seems to have been carried out (Table 6) but it isn’t described.

Overall the statistics section needs to explain every statistical model and method that was employed!

Response: Thanks for the reviewer’s comments. We have further described the statistical analysis section as follows:

The multivariate regression model was used to determine odds ratios (ORs) and 95% confidence intervals (CIs) for predicting in-hospital mortality. Variables for adjustment in the multivariate regression model were selected based on prior rationale and known associated risk factors. For in-hospital mortality (IHM), the crude model represents univariable analysis. Model A was adjusted for age, sex, blood pressure, and length of hospital stay. Model B included adjustments for age, NIHSS score, and mRS score. Additionally, model C was further adjusted for age, hypertension, and random blood glucose levels to evaluate novel glucose markers and their association with fatal outcomes. Additional information was added in the materials and methods section in revised manuscript (line 121-130).

What was the primary outcome? Death or severity? What were the power calculations?

Was their adjustment of p-value for multiple testing? The authors are basically testing several variables against a several outcomes

Response: The primary outcome of this study was in-hospital mortality. Additionally, we aimed to investigate the association of these biomarkers with unfavorable outcomes, including severity, complications, and death. We have revised the manuscript to improve clarity.

Results:

Overall, the number of figures and tables is too much and some of the data is redundant. This needs to be re-designed to avoid redundancies.

Table 1 – Include ischemic time if available. This will be one of the main drivers of mortality

Response: Thanks for the reviewer’s suggestion. The ischemic time was not available for collection; therefore, we incorporated this potential analysis into the limitation of this study in line 410-411 in this revised manuscript.

Table1 – only shows survivors vs. non-survivors. A table showing clinical characteristics of the so called severe and mild groups is also required

Response: We appreciate the reviewer’s kind suggestions. The data for severe and mild groups were not available for record. We have addressed this limitation in the manuscript and will consider it in future studies.

What were the main causes of death?

Response: In this study, AIS patients with pneumonia may be a major cause of in-hospital mortality.

Line 152 – how can glucose ratios be used for the diagnosis of cerebrovascular disease? Rephrase.

Response: We appreciate this suggestion and have rephrased it as “for the prognosis of various condition, including cerebrovascular disease” in line 172 in the revised manuscript.

Figure 1: why is survived vs. dead included here for GAR although it is already shown in Table 3?

Response: Thank you for the reviewer’s comment. We have revised Figure for comparing mild and severe cases of GAR.

Figure 2. How does this figure help in interpreting the data? There are already 2 significant differences in FPG. SHR and GAGR only show 3 signs. differences. what is the point of showing this?

Response: We are grateful for the reviewer’s comments. This figure illustrates the different levels of FPG and novel glucose markers in AIS patients with complications. Our results demonstrate that these markers are associated with complications, including pneumonia and sepsis, which may contribute to severity and in-hospital mortality. The manuscript text has been revised to enhance the clarity of the result in this revised manuscript in line 193-200.

Line 205-213: I cannot interpret this, as there is no definition of stroke severity. You could analyze correlation with NIHSS severity or some other objective parameter (e.g., extent of infarcted area on imaging). Furthermore, the AUC is very low.

Response: Severe cases, defined by an NIHSS score >5 at baseline, exhibited various clinical manifestations, including early neurological deterioration and the need for appropriate management, such as thrombolytic agents (e.g., rtPA), oxygen support, endovascular treatments, and transfer to the intensive care unit. However, delayed prediction, especially in clinical settings with limited medical resources, may contribute to in-hospital mortality. The manuscript text has been revised to enhance the clarity of the result in this revised manuscript in line 221-226.

Table 4: is this simple logistic regression?

Why are no ROC curves provided for multivariate analysis? This is the more interesting?

Response: Table 4 presents the results of the simple logistic regression. Our study aimed to investigate the potential ability of the novel glucose-related markers to predict severity and in-hospital mortality. We thus performed the ROC curve analysis for these glucose-related markers, but not for multivariate analysis. We have noted this as a limitation, considering it for further research in line 410-415 in this revised manuscript.

table 5: why would you use glucose parameters to predict the necessity of endovascular therapy? The decision is usually based on clinical and radiological parameters.

Response: We thank the reviewer for this suggestion. Clinical and radiological examinations are the primary criteria for decision-making in the appropriate management of endovascular therapy. However, our study may provide valuable data on laboratory markers as supportive tools for clinical management, especially in settings with limited resources. We have provided more information in line 285-289 in the current manuscript.

Table 6. Why does none of the models contain the diagnosis of diabetes and obesity, which will most likely be strong confounders?

Why are no further parameters of the multivariate logistic regression shown, e.g. ROC curves for the more promising quotients?

Moreover, GAR does not do so well in multivariate model A, so I am not so sure about an improved prognostic value compared to the other parameters.

Response: We would like to thank the reviewer for their valuable suggestions. We have considered these important points. As the data was not recorded, we have acknowledged this gap and limitation in our study. However, our study remains useful in illustrating alternative parameters associated with in-hospital mortality in these patients. Furthermore, we will take this key point into consideration for future studies.

Conclusion

The first sentence is far too bold given the small sample size, inherent bias etc.

Replace the first sentence, for example, with the following: “This study highlights the potential prognostic value of SHR and GAR in patients with AIS as these parameters demonstrated an association with in-hospital mortality.” The rest of the conclusion is fine.

Response: We thank the reviewer for their thoughtful review of our work. We have revised the conclusion in the current manuscript to ensure clarity without overstating the findings.

Response: We have named the files according to PLOS ONE's style requirements.

2. We note that there is identifying data in the Supporting Information file

Response: In the revised manuscript, we have removed the key data that should not be shared eg. Age in the supplementary file.

---

## [Decision Letter · Decision Letter 1]

14 Apr 2025

PONE-D-25-01318R1Evaluation of novel glucose-related blood biomarkers for predicting in-hospital mortality

in patients with acute ischemic strokePLOS ONE

Dear Dr. Angkasekwinai,

Thank you for submitting your manuscript to PLOS ONE. After careful consideration, we feel that it has merit but does not fully meet PLOS ONE’s publication criteria as it currently stands. Therefore, we invite you to submit a revised version of the manuscript that addresses the points raised during the review process.

We look forward to receiving your revised manuscript.

Kind regards,

Atakan Orscelik

Academic Editor

PLOS ONE

Journal Requirements:

Reviewers' comments:

Reviewer's Responses to Questions

**Comments to the Author**

1. If the authors have adequately addressed your comments raised in a previous round of review and you feel that this manuscript is now acceptable for publication, you may indicate that here to bypass the “Comments to the Author” section, enter your conflict of interest statement in the “Confidential to Editor” section, and submit your "Accept" recommendation.

Reviewer #1: All comments have been addressed

Reviewer #2: All comments have been addressed

2. Is the manuscript technically sound, and do the data support the conclusions?

Reviewer #1: Yes

Reviewer #2: Yes

3. Has the statistical analysis been performed appropriately and rigorously? 

Reviewer #1: Yes

Reviewer #2: Yes

4. Have the authors made all data underlying the findings in their manuscript fully available?

Reviewer #1: Yes

Reviewer #2: Yes

5. Is the manuscript presented in an intelligible fashion and written in standard English?

Reviewer #1: Yes

Reviewer #2: Yes

6. Review Comments to the Author

Reviewer #1: Dear Author

The revised manuscript are adressed properly with reviewer comments. So now the manuscript is ready to accept and publish

Reviewer #2: Thank you for providing the revised comments. The bias inherent to the study are mostly transparent and the methods are now clearer. I agree with the revised conclusion

The lack of including diabetes, a strong confounder, in the multivariate analysis needs to be mentioned explicitly in the limitations section.

7. PLOS authors have the option to publish the peer review history of their article (what does this mean? ). If published, this will include your full peer review and any attached files.

**Do you want your identity to be public for this peer review?** For information about this choice, including consent withdrawal, please see our Privacy Policy .

Reviewer #1: No

Reviewer #2: No

---

## [Author Response · Author response to Decision Letter 1]

21 Apr 2025

Review Comments to the Author

Reviewer #1:

Dear Author

The revised manuscript is addressed properly with reviewer comments. So now the manuscript is ready to accept and publish

Response: We warmly thank the reviewer for the thoughtful review and kind remarks regarding our work.

Reviewer #2:

Thank you for providing the revised comments. The biases inherent to the study are mostly transparent and the methods are now clearer. I agree with the revised conclusion

The lack of including diabetes, a strong confounder, in the multivariate analysis needs to be mentioned explicitly in the limitations section.

Response: We truly appreciate the reviewer’s insightful comments. The manuscript has been revised accordingly, and the suggested sentences have been added in the discussion at Line 414-415.

Response: We have verified all 39 references to ensure that none of them have been retracted.

---

## [Decision Letter · Decision Letter 2]

23 Apr 2025

Evaluation of novel glucose-related blood biomarkers for predicting in-hospital mortality

in patients with acute ischemic stroke

PONE-D-25-01318R2

Dear Dr. Angkasekwinai,

We’re pleased to inform you that your manuscript has been judged scientifically suitable for publication and will be formally accepted for publication once it meets all outstanding technical requirements.

Kind regards,

Atakan Orscelik

Academic Editor

PLOS ONE

Additional Editor Comments (optional):

Reviewers' comments:

Reviewer's Responses to Questions

**Comments to the Author**

1. If the authors have adequately addressed your comments raised in a previous round of review and you feel that this manuscript is now acceptable for publication, you may indicate that here to bypass the “Comments to the Author” section, enter your conflict of interest statement in the “Confidential to Editor” section, and submit your "Accept" recommendation.

Reviewer #2: All comments have been addressed

2. Is the manuscript technically sound, and do the data support the conclusions?

Reviewer #2: Yes

3. Has the statistical analysis been performed appropriately and rigorously? 

Reviewer #2: Yes

4. Have the authors made all data underlying the findings in their manuscript fully available?

Reviewer #2: Yes

5. Is the manuscript presented in an intelligible fashion and written in standard English?

Reviewer #2: Yes

6. Review Comments to the Author

Reviewer #2: The authors have addressed all my points. I believe the limitations and conclusions now give a balanced picture.

7. PLOS authors have the option to publish the peer review history of their article (what does this mean? ). If published, this will include your full peer review and any attached files.

**Do you want your identity to be public for this peer review?** For information about this choice, including consent withdrawal, please see our Privacy Policy .

Reviewer #2: No

---

## [Editor Report · Acceptance letter]

PONE-D-25-01318R2

PLOS ONE

Dear Dr. Angkasekwinai,

I'm pleased to inform you that your manuscript has been deemed suitable for publication in PLOS ONE. Congratulations! Your manuscript is now being handed over to our production team.

Kind regards,

on behalf of

Dr. Atakan Orscelik

Academic Editor

PLOS ONE